# Examining attitudes about the virtual workplace: Associations between zoom fatigue, impression management, and virtual meeting adoption intent

Chaeyun Lim[1]*, Rabindra Ratan[1], Maxwell Foxman[2], David Beyea[3], David Jeong[4], Alex P. Leith[5]

1 Michigan State University, East Lansing, Michigan, United States of America, 2 University of Oregon, Eugene, Oregon, United States of America, 3 University of Wisconsin-Whitewater, Whitewater, Wisconsin, United States of America, 4 Santa Clara University, Santa Clara, California, United States of America, 5 Southern Illinois University Edwardsville, Edwardsville, Illinois, United States of America

* limchae1@msu.edu

**Data Availability Statement:** All relevant data are within the manuscript and its Supporting information files.

## Abstract

Impression management is a crucial tactic within the workplace milieu. This study establishes a connection between impression management and the negative self-evaluation stemming from heightened self-monitoring during virtual meetings (VM), which manifests in the form of Zoom (VM) fatigue. We conducted a cross-sectional survey, by recruiting 2,448 U.S.-based workers. Our survey results revealed that facial appearance dissatisfaction is associated with VM fatigue, resulting in lower intention to adopt VM technologies due to decreased perceived usefulness of VM technologies. Furthermore, building upon the Uses and Gratification (UG) perspective and the assumptions of the Social Information Processing (SIP) theory and the Hyperpersonal Model, our findings illuminate that VM fatigue prompts the use of impression management behaviors by using VM features closely linked to dissatisfaction with one's facial appearance. The results suggest that utilization of impression management features in VMs is driven by needs related to facial appearance concerns, which is associated with impression management. This study extends the concept of impression management to VM environments in the workplace, underscoring the importance of addressing workers' needs and well-being to foster worker-friendly VM communication environments and promote VM acceptance. This study identifies external factors within the Technology Acceptance Model by integrating the UG perspective, the SIP theory, and the Hyperpersonal Model to understand the mechanisms underlying VM fatigue and adoption in the emerging virtual workplace.

## Introduction

The increasing reliance on virtual meetings (VMs) has led to a pervasive experience of VM fatigue, commonly referred to as "Zoom fatigue" [1]. This phenomenon has significant

**Funding:** Rabindra Ratan, Maxwell Foxman, David Beyea, Alex P. Leith have received funding from NSF. This material is based upon a study supported by the National Science Foundation under grant FW-HTF-R: Collaborative Research: Virtual Meeting Support for Enhanced Well-Being and Equity for Game Developers NSF Award Nos. 2128746, 2128803, 2128813, and 2128991 from SES Division. NSF URL: https://www.nsf.gov/ NSF did not play any role in the study design, data collection and analysis, decision to publish, or manuscript preparation.

**Competing interests:** The authors have declared that no competing interests exist.

implications for workplace productivity and individual well-being, as VM fatigue impedes effective engagement with interactions in VMs [2]. Notably, women and people of color report higher levels of VM fatigue, potentially stemming from facial dissatisfaction during self-view [2–4]. While disabling self-view has been suggested as an effective solution [5–7], it remains impractical for many users who prefer to self-monitor during meetings. Beyond disturbing workplace interactions and productivity, such negative experiences can create psychological barriers to adopting VM technologies [8], contributing to technology inequity in the workplace. Despite VM fatigue's critical role in shaping workplace interactions and digital inclusion in emerging virtual work environments, its effects on VM adoption—and the mechanisms linking facial appearance concerns, VM fatigue, and VM adoption—remain underexplored.

This study responds to the pressing need to understand the mechanism of VM fatigue and its consequences for virtual workplace technology inclusion. Specifically, we investigate impression management features–tools that enable users to adjust their self-video to manage their appearance. By exploring the relationships among facial appearance concerns, VM fatigue, and behavioral intentions toward VM technologies, the present research uncovers what motivates the use of these tools to manage appearance-related concerns and how such usage can promote equity, inclusion, and user satisfaction.

This work's theoretical contributions include extending the concept of impression management to encompass appearance-related concerns in VM contexts, with broader implications for equity and inclusion in remote work environments. The research also advances understanding of the psychological mechanisms underlying VM fatigue and its influence on technology adoption by drawing upon the Uses and Gratifications (UG) perspective, Computer-Mediated Communication (CMC) theories (i.e., the Social Information Processing (SIP) theory and the Hyperpersonal Model), and the Technology Acceptance Model (TAM). By connecting facial appearance dissatisfaction, appearance-related impression management behaviors via VM technologies, and VM fatigue, the study highlights actionable pathways for VM platforms to enhance user well-being and their adoption of the technology.

## Facial appearance dissatisfaction and VM

Excessive screen time, engagement with social media and selective self-presentation through modifying photos before posting them have long been associated with appearance dissatisfaction [9]. Similarly, the extended amount of time spent on VMs may exacerbate negative perceptions of self-image, as well as concerns of critical evaluation. There are studies that report that the number of people who seek cosmetic procedures has increased during the pandemic, and looking at themselves through cameras during VMs may be one contributing reason for this occurrence [10,11]. This phenomenon, known as Zoom Dysmorphia—referred to as feeling displeased with one's physical appearance, may potentially stem from excessive exposure to front-facing cameras, which distort self-appearance [10–12]. Excessive viewing of oneself on screens can lead to heightened over-focus, resulting in users being overly self-aware and negatively evaluating themselves [4,13,14]. This self-focus can then increase negative feelings by the individual, such as anxiety and depression [15]. Indeed, the COVID19 pandemic saw increased trends regarding facial appearance concerns [16]. The most frequently raised concerns of patients who saw dermatologists during the pandemic were facial appearances that are mainly exposed to others through the camera during VMs [10], and most patients cited the reason for their concerns as VMs.

The potential underlying mechanism of this unintended consequences of excessive self-viewing can be explained by the expanded gap between the ideal self and the actual self, which is manifested by over-scrutinization of their facial features caused by an excessive amount of

screen time [17,18]. Recent studies found that the amount of time spent on videoconferencing meetings was indirectly associated with appearance satisfaction because of the increased tendency to compare oneself to others [19,20]. Studies revealed that increased self-attention through mirror gazing can contribute to negative evaluation on self-appearance of a mixture of facial and body features [21–23]. In the context of videoconferencing meetings, however, individuals are boxed into a specific view of the self that keeps one's face within the camera frame [5]. Thus, the present study specifically seeks to understand the psychological mechanism of facial appearance dissatisfaction in the workplace VM context. To be specific, we posit that facial appearance dissatisfaction is positively associated with impression management behaviors in the VM context, through the mediating role of VM fatigue. Furthermore, building upon technology adoption model (TAM), the current study will investigate the potential impact of impression management behaviors in VMs on behavioral intention to adopt VM technologies for workplace meetings.

## Facial dissatisfaction and VM fatigue

Physical and mental exhaustion from VMs, VM (VM) fatigue—also called Zoom fatigue [1], has become a prevalent phenomenon since the increase in use of VM technologies for workplace meetings, mainly driven by the COVID-19 pandemic. A substantial body of literature has investigated the underlying mechanism of VM fatigue [5,24]. In the social media context, concerns over self-impression showed a positive relationship with fatigue, and this relationship was stronger for women than men [25]. Similarly, facial appearance dissatisfaction can cause significant distress in VMs as people can overly focus on their own and others' facial appearances. Individuals unhappy with their facial appearance might experience intensified distress when seeing themselves on video during VMs, leading to a vicious cycle of negative self-evaluation. These negative responses can also create a challenging loop of unpleasant feelings toward VMs, adversely influencing individual well-being and workplace interactions.

Dissatisfaction with one's facial appearance is known to be linked to VM fatigue [3], supporting the reasoning that one of the causes of VM fatigue might be psychological distress induced by excessive focus on facial appearance [5]. There is empirical evidence that camera use increased VM fatigue [26], indicating that facial dissatisfaction may potentially play a substantial role in triggering VM fatigue. Given that many teams and organizations encourage people to turn on cameras to signal their presence during VMs [27] and people often want to monitor how they appear through their front-facing cameras [28], facial appearance dissatisfaction is likely to entail VM fatigue. The present study will replicate the previous findings of the impact of facial appearance dissatisfaction in VM fatigue and expand this relationship as a mediating path toward impression management behaviors by using VM features.

**H1:** The more people are dissatisfied with their facial appearance, the more people will suffer from Zoom (VM) fatigue.

## Facial dissatisfaction and impression management behaviors in VMs

Impression management, referred to as the control of information about oneself to shape others' perceptions [29], influences not only how individuals are perceived by others but also satisfaction with meeting interactions [30]. Scholarship on computer-mediated communication suggests that individuals tend to prefer communication channels that afford multiple opportunities to control their self-presentation, enabling more effective self-impression management during communication [31]. Altering one's appearance to appear desirable in social situations

is a prominent form of impression management, also known as selective self-presentation [32–34]. For example, people worry about appearing older and being regarded as unattractive in diverse social interactions [35]. This negative appraisal can result in unethical hiring practices and acknowledgment in the workplace [36–38]. This concern over appearance and its possible consequences can lead individuals to perform strategic self-presentation to shape impressions [39–44]. To create desirable self-images, people seek cosmetic procedures or products that boost their physical appearance [45–47]. Furthermore, clothing is an essential aspect of impression management, affecting credibility, reliability, and competence, especially in the workplace context [48–52].

Such psychological and social phenomena of impression management—altering one's appearance to create desirable impressions—are mirrored in online environments. When people post pictures of themselves online, they selectively self-present by choosing what they perceive as their best pictures [53,54]. When it comes to VMs, impression management behaviors can appear in the form of specific feature use (e.g., filtering, touch-up, and avatars), which allows users to strategically present themselves by controlling their outward presentation. For instance, existing literature suggests that the underlying motivation for avatar customization in virtual environments may be strategic self-presentation [55]. Given that VMs in the workplace entail complex team dynamics that require constantly adjusting behaviors, individuals may engage in impression management by using available VM features to cultivate favorable self-perception in meeting contexts.

Concerns regarding one's own facial appearance in VMs, intensified by monitoring self-videos, may lead to an increased inclination toward impression management tactics by utilizing available VM features. As physical attractiveness can be a substantial factor influencing how others perceive an individual's favorable qualities in organizational settings, individuals may become concerned about the impressions they make based on their physical appearance [56,57]. Indeed, impression management is a fundamental behavioral tactic commonly observed in meeting interactions [58]. People tend to focus on negative information about themselves captured by the camera [59]. Exposure to self-mirror images during professional VMs may elicit critical self-impression evaluation [60], especially for individuals who already are uncomfortable with their own facial appearance. In the physical world, individuals having higher facial appearance concerns often consider cosmetic surgery [61]. In online environments, visual self-representations, such as social media profile pictures, serve as tools for impression management [62]. A body of literature revealed that the greater the dissatisfaction with one's appearance, the more likely individuals are to alter their photos to appear more attractive [63–68]. Similarly, dissatisfaction with one's own facial appearance can prompt individual desires for impression management by using available features in the VM context.

According to Social Information Processing (SIP) theory, communication participants will take adaptive advantage of the cues and features within a digital environment to make up for those social cues that are unavailable in the environment and thereby achieve their communication needs [69,70]. While SIP theory is designed for text-based computer-mediated communication, we can address this adaptation principle when examining impression management behaviors in VMs [71]. Following the framework that individuals adapt to the features and cues available to them, we can posit that individuals will use the features afforded by VM technologies to achieve their communication goals (i.e., giving positive impressions overcoming perceived negative facets of self-appearance) in this context. Considering the substantial role of nonverbal cues in signaling participants' states and attitudes in the VMs [72,73] where there are fewer available cues [5,71] compared to face-to-face meeting environments, it is likely that as individuals tend to utilize available VM features as cues to aid in managing self-impressions.

Walther [70] further developed the Hyperpersonal Model, proposing that CMC can transcend face-to-face communication—bringing people closer together than if they were communicating in person—by (in part) offering unique communication opportunities for selective-self presentation. The Hyperpersonal Model suggests that asynchronous CMC affords users more time to construct the information they wish to convey compared to face-to-face interactions, thereby enabling more desirable self-presentation in CMC environments. Although the Hyperpersonal Model was developed in asynchronous text-based CMC contexts, it is also applicable to VMs, which also provide opportunities for selective self-presentation between communication partners. For example, voice and video filters or touch-up features on VM platforms can function as tools to aid self-presentation. Recent literature suggests that individuals often make an effort to present positive, energetic facial expressions during VMs [74], oftenly by using video filters to make their facial expressions more visible [75]. A recent study revealed that individuals use AR facial filters for ideal self-presentation [76]. Additionally, evidence suggests that having greater self-appearance concerns—arising from excessive self-monitoring through self-video streaming in VMs—leads individuals to engage more with self-enhancement features during these meetings [77].

In sum, VM software offers features that can help users accomplish communication goals for impression management, especially in response to facial appearance concerns. Hence, we expect that facial appearance dissatisfaction leads to greater use of impression management tools in VM:

**H2:** The more people are dissatisfied with their own facial appearance, the more people engage with impression management behaviors by using features afforded by VM technologies.

### VM Fatigue and impression management

Despite the substantial role of impression management documented in workplace interactions [78], it is unclear how it manifests in VM contexts. The Uses and Gratifications (UG) perspective suggests that individuals proactively select media technologies to fulfill their needs and desires [79]. Although UG has been extensively applied to traditional media, providing explanations on socio psychological motivations for using specific media technologies, its application to VM context has been limited [1]. Given the unique gratifications offered by different media technologies [80], employing various impression management tools (e.g., avatars, touch-ups, and video/audio filters) in VMs could potentially mitigate VM fatigue. For instance, filters and avatars may serve as a buffer layer that affords self-view occlusions, thereby easing the negative impacts of excessive self-awareness. A recent study showed that touch-up features can enhance facial appearances by alternating perceived flaws of one's face, contributing to improved self-image [76]. Similarly, in the physical world, self-image enhancement by deliberate clothing decisions and cosmetics use can shape positive self-perception [47]. This resonates with the UG perspective, suggesting that motivations such as self-expression [81], concentration [82], and relaxation [83,84] play a crucial role in media technology selection. For instance, a recent study found that individuals experiencing higher levels of VM fatigue in their current workplace show a greater willingness to adopt avatar customization behaviors in future metaverse workplaces [85]. Individuals suffering from VM fatigue may thus be inclined to use available impression management features to alleviate VM fatigue by controlling their self-presentation in VM situations. Considered collectively, the present study proposes hypotheses concerning the positive link between VM fatigue and the engagement in impression management behaviors through VM technology features, as well as the mediating role of

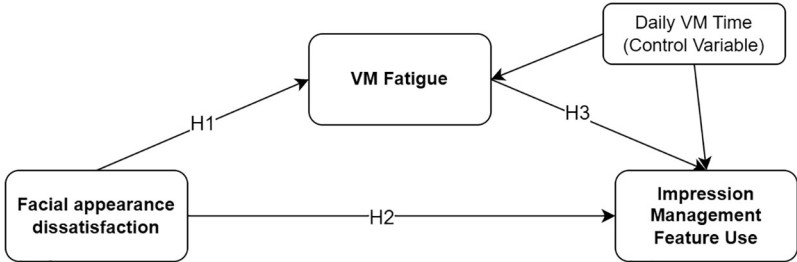

**Fig 1. Model 1: The hypothesized mediation model predicting usage of impression management feature.**

VM fatigue in the relationship between facial appearance dissatisfaction and impression management behaviors, as illustrated as Model 1 in Fig 1.

**H3:** The more people experience VM fatigue, the more they engage with impression management behaviors by using features afforded by VM technologies.

**H4:** VM fatigue mediates the relationship between facial appearance dissatisfaction and impression management feature use.

## VM fatigue and behavioral intention to adopt VM technology

To understand user acceptance and attitudes toward information and communication technologies, scholars have commonly applied the Technology Acceptance Model (TAM) [86], inspired by the Theory of Reasoned Action [87]. See Fig 2. According to TAM, the two substantial variables that affect the actual use of the technology through attitudes toward the technology are perceived usefulness (PU) and perceived ease of use (PEOU). PU is "the degree to which a person believes that using a particular system would enhance their job performance," while PEOU is "the degree to which a person believes that using a particular system would be free of effort" [86]. TAM assumes the existence of external factors in shaping PU and PEOU, such as technology characteristics and training programs for the technology use [86]. Building upon this approach, researchers have investigated various factors that link to PU and PEOU to develop more comprehensive knowledge about technology acceptance [88,89]. Researchers have explored determining factors of PU and PEOU in virtual learning and communication contexts such as usability, facilitating conditions, playfulness, system quality, enjoyment, and individual differences such as self-efficacy [90–95].

VM fatigue, as a negative emotion experienced during and after VMs, can be a hindrance to feeling that VM technology is effective and easy to use. Because VM fatigue entails general, emotional, and physical burdens [1], individuals who experience VM fatigue may find the meeting technology more difficult due to limited capacity to familiarize themselves with

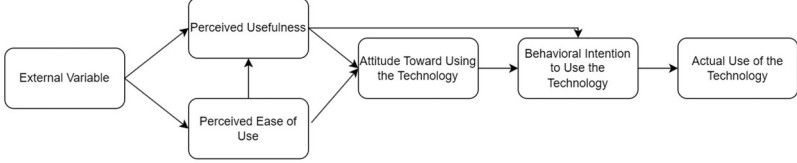

**Fig 2. TAM suggested by Davis (1989) [86].**

utilizing the VM technology. For instance, an individual who feels physically constrained, facing hindered body movement during videoconferencing meetings [5] may experience VM fatigue, potentially resulting in a decreased utilization of VM features. This could lead to a limited capacity to familiarize themselves with VM technologies. Furthermore, VM fatigue drains social/mental energy to interact with colleagues and to concentrate on tasks, affecting impaired performance. Workers who experience VM fatigue more may tend to perceive their videoconferencing platform as an unsuccessful workplace tool. An additional potential underlying mechanism for the impact of Zoom fatigue on PU and PEOU could be that the adverse psychological state might lead to a pessimistic assessment of the technology [96].

A construct similar to VM fatigue, for instance, technostress—the overload and tension induced by using novel technologies [97]—was found to be negatively associated with technology acceptance behaviors [98–102], potentially resulting in innovation resistance within organizations. Recent literature exploring the adoption of remote teaching technologies showed that technostress undermines PU and PEOU, leading to negative intentions to use the technology [103]. Similarly, anxiety about teleconferencing systems has been shown to reduce PU, thereby decreasing the actual use of these systems [104]. On the contrary, some research highlights the impact of positive experiences with VM technologies. One study found that the intention to adopt Zoom meetings was predicted by the perceived excitement of using the platform [105], suggesting that positive experiences with VM technologies can promote adoption. These findings suggest that while negative experiences with VM technologies can create psychological barriers and hinder adoption, positive experiences can foster acceptance and continued use of VM technologies.

In contrast, a recent study of online educational technology adoption found that Zoom (VM) fatigue was not significantly associated with actual use [106]. However, this research only investigated the direct effect of VM fatigue on technological use. We still expect substantial relationships between VM fatigue and adoption indirectly, through mediating effects of PU and PEOU, respectively. We propose the following hypotheses to augment the body of evidence suggesting that the subjective assessment of technology hinges on individual experience, focusing on VM fatigue as the underlying determinant of VM technology adoption. For succinctness, we explore only the intention to adopt the VM technology as the final outcome variable without further exploration of attitudes toward use as originally suggested by Davis (1989) [86]. Also, we expect a path from PEOU to PU, which is well-established in TAM literature, but is not a focus of this paper. Instead, we focus on combined serial mediation effects. The hypothesized model is shown as Model 2 in Fig 3.

**H5:** The less people suffer from VM fatigue, the more people think the VM technology is useful.

**H6:** The less people suffer from VM fatigue, the more people think the VM technology is easy to use.

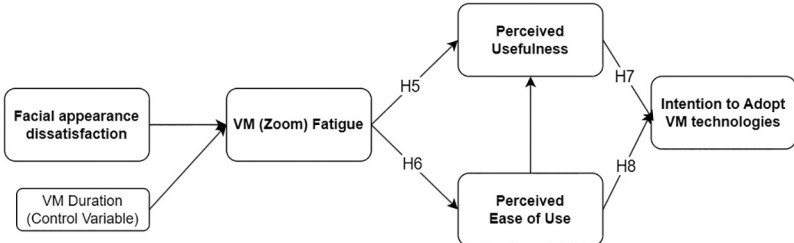

**Fig 3. Model 2: Hypothesized mediation model incorporating TAM.**

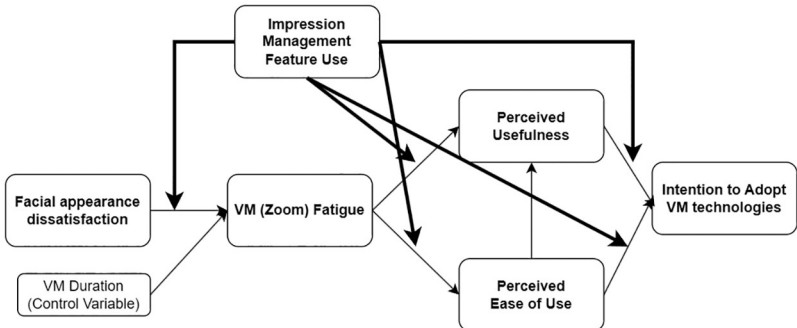

**Fig 4. Model 3: The potential moderation model with the role of usage of impression management features in VM.**

**H7:** The more people think the VM technology is useful, the greater people have intention to adopt the VM technology.

**H8:** The more people think the VM technology is easy to use, the greater people have intention to adopt the VM technology.

**H9:** There is a serial mediation effect of VM fatigue, perceived ease of use, and perceived usefulness in the relationship between facial appearance dissatisfaction and intention to adopt the VM technology.

We further explore the possible moderating effect of impression management feature use. Studies suggest that external factors such as uncertainty reduction [107], support [108], experiences with the technology [109,110] moderate the impact on information technology adoption. Building upon the notion, we expect a potential moderating role of usage of impression management features in influencing behavioral intention. Namely, the usage of technology features can serve as both an experience with the technology [109,110] and a reflection of technology awareness [111,112], which is a well-known moderator in the TAM context. Individuals engaging more with impression management behaviors in VMs might exhibit weaker associations between variables. For example, the negative relationship between facial appearance dissatisfaction and VM fatigue could be less pronounced among those who frequently use impression management features. Conversely, it is also feasible that those utilizing such features more extensively may be ones already facing the adverse effects of appearance concerns and VM fatigue, making the effect more pronounced. Extending H3, which discusses the relationship between the use of impression management features and VM fatigue, we aim to investigate the moderating role of impression management feature use. See Fig 4.

**RQ1**: Does usage of impression management features moderate the effects of facial appearance dissatisfaction, VM fatigue, PU, and PEOU on intention to adopt the VM technology?

## Methods

### Ethics statement

We conducted an anonymous survey, reviewed and approved by the Human Research Protection Program at [university name anonymized]. This survey was part of a large project to

understand remote workers' VM experiences and well-being. Participants were asked to provide written informed consent prior to entering the survey. Once they provided written consent, they were able to proceed with the survey. This study did not include any minors. The survey was conducted from February 7, 2023 to April 6, 2023.

## Participants

We recruited 2,448 U.S.-based workers to participate in the single cross-sectional 15-minute survey through a research panel provided by Qualtrics. The company restricted the sample to include professional, technical, or scientific workers who worked remotely at least sometimes and who participated regularly in virtual meetings for work. The sample was intentionally stratified by two gender categories (men and women) and four racial categories (African American, Asian, Hispanic or Latino, and White)—chosen as representative of the most prominent groups in the sample—with approximately even participation among the eight subgroups in order to facilitate statistical comparisons. The survey questions were randomly distributed to the overall sample of 2448 participants to reduce the duration of the survey and to minimize measurement errors caused by response fatigue.

The sample for Model 1 analysis was **453** (the number of participants who answered to facial appearance dissatisfaction, VM fatigue, impression management feature use, and daily VM duration, Women: 227, Men: 226). The sample for Model 2 was **267** (the number of participants who answered to facial appearance dissatisfaction, VM fatigue, TAM variables, and daily VM duration, Women: 139, Men: 128). The sample for Model 3 analysis was **228** (the number of participants who answered to facial appearance dissatisfaction, VM fatigue, impression management feature use, daily VM duration, and TAM measures). Of the total participants included in the analyses for Model 3 ($n$ = 228), 117 reported they are women, 111 men. Participants' age ranged from 18 to 81 ($M$ = 41.66, $SD$ = 13.51). Their tenure at the workplace ranged from 1 year to 21 years ($M$ = 9.5, $SD$ = 6.21). Regarding race, 56 reported being African American, 52 Asian, 73 Hispanic, and 47 White. Of the 267 participants for Model 2, 39 were excluded from the analyses for Models 1 and 3 because they indicated that impression management features were not applicable to the virtual meeting platform they primarily used. All participants included in Model 3 were included in the Model 1 analysis. As this survey was a single cross-sectional study, all responses for the study's models were collected at the same time.

## Measures

All response options for measures are 5 Likert-type scales ranging from 1, "Not at all" to 5 "Extremely," unless we present a different option. Participants were asked to answer the most commonly used VM platform they use at the beginning of the survey among the possible response options ranging from Zoom, Google Meets, Gather.town, Facetime, etc. Once they answer this question, their response (e.g., Zoom) was tailored to ask the following questions. The questions for each measure included the phrase 'the VM platform you primarily use,' which was replaced by each participant's response (e.g., Zoom or Gather.town). To prevent common method bias, we ensured participant anonymity before introducing the survey questions. Additionally, we included unrelated questions that were not tied to our variables of interest to minimize the potential for consistent response patterns caused by priming or consistency effects.

**Facial appearance dissatisfaction.** Participants' negative perceptions on their facial appearances were measured by using four items by adapting one dimension of negative physical self scale [113]. The items included: "I am depressed about how my face looks," "If it is

possible, I will change the way my face looks," "If there is some way I can improve my face, I will keep trying to do it," and "I am ashamed about my facial appearance." (Cronbach's $\alpha$ = .88, $M$ = 2.00, $SD$ = 1.02).

**Impression management in VMs.**   To measure impression management behavior by using VM features, we created eight items about possible and salient features for impression management in the VM environment. As part of a study to understand remote workers' VM behavior, the original survey questionnaire included 20 more items about feature use (28 items in total). Participants were asked to answer the question "Thinking about your meetings for work on 'the VM platform you primarily use,' how often do you think the following statements are true?" The response options ranged from 1, "Never" to 5, "Always." We also included "not applicable in [the platform I primarily use]," as some VM platforms may not afford the same features. We excluded the cases that chose the "not applicable" option. We carved out four impression management features based upon EFA and the reliability analysis, which is beyond the scope of the current study. The parallel analysis [114,115] suggested seven factors are the appropriate number for the 28 items. The EFA with seven factors with promax rotation [116] suggested that six items among the original eight items were significantly loaded together (above factor loading $\lambda$ of .30).

However, we, the authors discussed that following two items out of the six items were somewhat less relevant to our conceptual definition of impression management behaviors (i.e., altering one's appearance to create desirable impressions): "I turn off my self-video so I cannot see myself, but others can still see me ($\lambda$ = .58)" and "I usually turn off my self-video for everyone so others only see a picture of me or my name ($\lambda$ = .39)," with relatively low factor loadings than other four items. As a result, the four items that we included as impression management behaviors in VMs were the following: "I use touch-up to enhance my self video ($\lambda$ = .66)," "I use video filters (image overlays that partially cover my self-video image) to change how I appear to others ($\lambda$ = .67)," "I use avatars (that fully cover my video image of my face/body) to change how I appear to others ($\lambda$ = .80)," and "I use filters for my voice so others cannot hear my own voice ($\lambda$ = .89)." (Cronbach's $\alpha$ = .86, $M$ = 1.91, $SD$ = .99).

**VM fatigue.**   VM fatigue was measured by fifteen items adapted by [1]'s Zoom Exhaustion and Fatigue (ZEF) scale. The measure includes five dimensions: general, visual, social, motivational, emotional fatigue induced by VMs. Same items include "How much do you tend to avoid social situations after [primary VM platform name] meetings" and "How much do you dread having to do things after [primary VM platform name] meetings." (Cronbach's $\alpha$ = .97, $M$ = 2.10, $SD$ = .96).

**Perceived usefulness.**   Perceived usefulness of the VM platform each participant commonly uses for work meetings was measured by adapting three items from [117]. The items included "I find 'the VM platform' useful in my life," "Using [primary VM platform name] enables me to accomplish tasks more quickly," "Using [primary VM platform name] increases my productivity." (Cronbach's $\alpha$ = .87, $M$ = 3.83, $SD$ = .87).

**Perceived ease of use.**   Perceived ease of use of the VM platform each participant commonly uses for work meetings was measured by adapting three items from [117]. The items included "It is easy to become skillful at using [primary VM platform name]" "I find [the VM platform I primarily use] easy to use," and "Learning to operate [primary VM platform name] was easy for me." (Cronbach's $\alpha$ = .88, $M$ = 4.03, $SD$ = .74).

**Intention to use the VM technology.**   Since the goal of the study is to investigate individual behavioral intentions toward the VM technology they mainly use for work meetings. The three items were used to measure this construct. The items included "I intend to use [primary VM platform name] for work meetings in the future," "I will likely use [primary VM platform

name] for work meetings in the future," and "I expect that I will use [primary VM platform name] for my work meetings in the future." (Cronbach's $\alpha$ = .89, $M$ = 4.12, $SD$ = .75).

**Daily VM duration.** It is more likely that individuals will experience more VM fatigue and use the available impression management features more once they become familiar with the software and technology. As an individual invests more time in participating in VMs, they will become more familiar with the available features and realize their usefulness for effective communication and strategic self-presentation. To control for the duration of average daily time spent on VMs to predict VM fatigue and impression management behaviors in VM in our statistical analyses, participants were asked to answer "In an average work day, how many hours do you tend to spend in work-related video meetings on [primary VM platform name]?" The response option included twenty one response options; "0" "less than 1 hour," "1," "2," to "19+ hours." The median value was 4 hours.

## Data analysis

To investigate the hypotheses and research questions, we conducted a structural equation modeling (SEM) analysis. The main SEM analysis was conducted by using the R package, 'lavaan.' We adopted Maximum Likelihood (ML) estimation as normality assumption for all endogenous variables was satisfied by showing skewness values ranged from -.70 to .87, kurtosis values ranged from 2.53 to 3.39, within the recommended values for SEM (a range of -3 to 3 for skewness and -10 to 10 for kurtosis) [118–120]. Prior to the analyses, we conducted Harman's Single-Factor Test to investigate the existence of significant common method bias. The results indicated that 39% of variance was explained by the single factor model, below the commonly accepted threshold of 50% [121]. This suggests that our data were not significantly affected by common method bias.

## Results

### Correlation between the variables of interest

Prior to the main SEM analysis, we conducted correlation tests to examine bivariate relationships with the variables of interest. The exogenous variable, facial appearance dissatisfaction showed positive correlations with impression management feature use, VM fatigue, and daily VM duration. Impression management feature use was positively associated with VM fatigue. VM fatigue was negatively correlated with PU, PEOU, and intention to use the VM technology. Daily VM duration had positive correlations with VM fatigue and impression management feature use, thus we included it as a control variable in our further analyses, as planned. All TAM variables were positively correlated with each other. See Table 1.

### Model 1 results

To investigate H1-H4 (Model 1), we conducted a SEM analysis. The hypothesized mediation model resulted in a good model to the data: $\chi 2(248)$ = 777.509, $p < .001$ RMSEA = 0.069 (90% CI, [0.063, 0.074]), $p$ = 0.000; CFI = 0.937; TLI = 0.930; SRMR = 0.063 [122–124].

Our model showed that facial appearance dissatisfaction was significantly and positively related to VM fatigue ($b$ = .554, SE = .048, $\beta$ = .553, $p < .001$), when controlling for daily VM duration ($b$ = .037, SE = .014, $\beta$ = .107, $p$ = .008). H1 was supported. Facial appearance dissatisfaction was significantly and positively associated with impression management feature use ($b$ = .372, SE = .059, $\beta$ = .370, $p < .001$), when controlling for daily VM duration ($b$ = .087, SE = .015, $\beta$ = .249, $p < .001$). H2 was supported. VM fatigue was significantly and positively associated with impression management feature use ($b$ = .237, SE = .055, $\beta$ = .236, $p < .001$), when

**Table 1. Correlation coefficients.**

|  | 1 | 2 | 3 | 4 | 5 | 6 | 7 | 8 |
|---|---|---|---|---|---|---|---|---|
| 1. Gender |  |  |  |  |  |  |  |  |
| 2. Tenure (Years of Job experience) | -.12 |  |  |  |  |  |  |  |
| 3. Daily VM duration | .04 | -.13 |  |  |  |  |  |  |
| 4. Impression management feature use | .05 | -.16* | .41*** |  |  |  |  |  |
| 5. Facial Appearance Dissatisfaction | .14* | -.18** | .23*** | .60*** |  |  |  |  |
| 6. VM Fatigue | .17* | -.26*** | .22*** | .48*** | .64*** |  |  |  |
| 7. Perceived Usefulness (PU) | -.03 | .02 | .18** | .08 | -.05 | -.26*** |  |  |
| 8. Perceived Ease of Use (PEOU) | -.11 | -.09 | .05 | -.01 | -.18* | -.19** | .51*** |  |
| 9. Intention to use the VM technology | .00 | .02 | .05 | -.12 | -.19* | -.32*** | .58*** | .42*** |

*Note.*

* p < .05,

** p < .01,

*** p < .001,

N = 228.

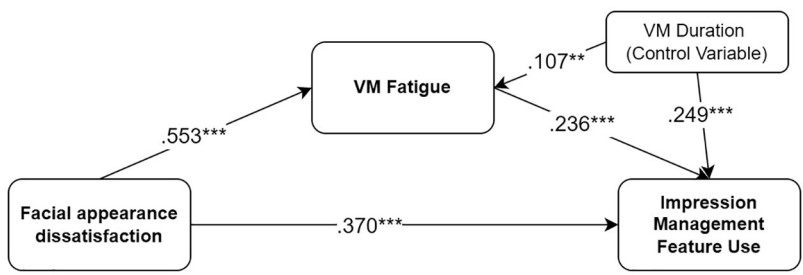

**Fig 5. Model 1 examined.**

controlling for daily VM duration. H3 was supported. The results revealed that there was a significant and positive mediating role of VM fatigue in the relationship between facial appearance dissatisfaction and impression management feature use ($b = .131$, SE = .032, $\beta = .120$, $p < .001$). Hence, H4 was supported. See Fig 5.

## Model 2 results

To investigate H5-H8 (Model 2), we also conducted a SEM analysis. The model resulted in a good model fit to the data: $\chi 2(367) = 721.465$, $p < .001$ RMSEA = 0.060 (90% CI, [0.054, 0.067]), $p = 0.006$; CFI = 0.946; TLI = 0.940; SRMR = 0.053 [122–124].

VM fatigue was significantly and negatively related to PU ($b = -.239$, SE = .063, $\beta = -.291$, $p < .001$), supporting H5. The relationship between VM fatigue and PEOU was not statistically significant ($b = -.047$, SE = .069, $\beta = -.062$, $p = .493$). Thus, we could not support H6. PU was significantly and positively related to intention to adopt the VM technology ($b = .409$, SE = .073, $\beta = .459$, $p < .001$), supporting H7. As suggested by the earlier TAM, PEOU and PU were significantly and positively related to each other ($b = .668$, SE = .072, $\beta = .618$, $p < .001$). PEOU and intention to adopt the VM technology were significantly and positively related ($b = .194$, SE = .074, $\beta = .201$, $p = .009$), supporting H8. The results indicated that there was a serial

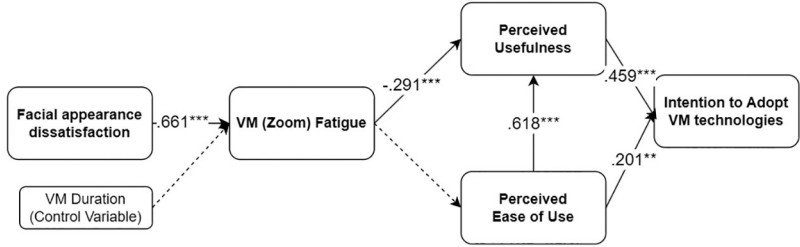

**Fig 6. Model 2 examined.**

mediation effect of VM fatigue and PU in the association between facial appearance dissatisfaction and intention to adopt the VM technology ($b$ = -.062, SE = .020, $\beta$ = -.088, $p$ = .002). However, the results revealed that the serial mediation path of VM fatigue, PEOU, and PU was not significant ($b$ = -.004, SE = .006, $\beta$ = -.005, $p$ = .507). See Fig 6.

## Model 3 results

To answer RQ1, we conducted a multigroup SEM. We divided the sample into two groups, based upon the median value (1.5) of impression management feature use (High use group: $n$ = 111, Less use group: $n$ = 117). The model fit for each group resulted in a good model, meaning that we can proceed with the model comparison.

For the **high** use group, facial appearance concerns were significantly and positively associated with VM fatigue ($b$ = .633, SE = .084, $\beta$ = .764, $p$ < .001). VM fatigue was significantly and negatively related to PU ($b$ = -.270, SE = .121, $\beta$ = -.358, $p$ = .026). The relationship between VM fatigue and PEOU was not statistically significant ($b$ = -.031, SE = .121, $\beta$ = -.045, $p$ = .799). PU was significantly and positively related to intention to adopt the VM technology ($b$ = .540, SE = .129, $\beta$ = .597, $p$ < .001). PEOU and PU were significantly and positively related to each other ($b$ = .667, SE = .144, $\beta$ = .604, $p$ < .001). The results showed that the serial mediation effect of VM fatigue and PU in the association between facial appearance dissatisfaction and intention to adopt the VM technology was marginally significant ($b$ = -.092, SE = .047, $\beta$ = -.163, $p$ = .052). The relationship between PEOU and intention to adopt the VM technology was not significant ($b$ = .203, SE = .131, $\beta$ = .203, $p$ = .122). The serial mediation path of VM fatigue, PEOU, and PU was not significant ($b$ = -.003, SE = .01, $\beta$ = -.004, $p$ = .801).

For the **low** use group, facial appearance concerns were significantly and positively associated with VM fatigue ($b$ = .559, SE = .114, $\beta$ = .456, $p$ < .001). VM fatigue was significantly and negatively related to PU ($b$ = -.288, SE = .082, $\beta$ = -.321, $p$ < .001). The relationship between VM fatigue and PEOU was not statistically significant (b = -.095, SE = .091, $\beta$ = -.115, $p$ = .294). PU was significantly and positively related to intention to adopt the VM technology ($b$ = .512, SE = .104, $\beta$ = .592, $p$ < .001). PEOU and PU were significantly and positively related to each other ($b$ = .606, SE = .095, $\beta$ = .560, $p$ < .001). The results showed that the serial mediation effect of VM fatigue and PU in the association between facial appearance dissatisfaction and intention to adopt the VM technology was statistically significant ($b$ = -.082, SE = .033, $\beta$ = -.087, $p$ = .012). The relationship between PEOU and intention to adopt the VM technology was not significant ($b$ = .039, SE = .098, $\beta$ = .041, $p$ = .693). The serial mediation path of VM fatigue, PEOU, and PU was not significant ($b$ = -.001, SE = .003, $\beta$ = -.001, $p$ = .712).

To statistically compare the coefficients of the two groups, we conducted Wald tests. We found no significant differences between all coefficients of the two groups, suggesting that the

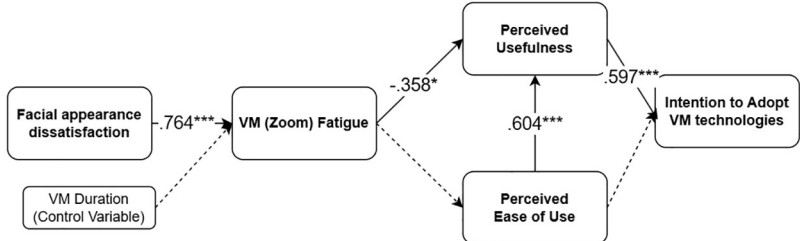

**Fig 7. Model 3 examined for the *high* impression management feature use group.**

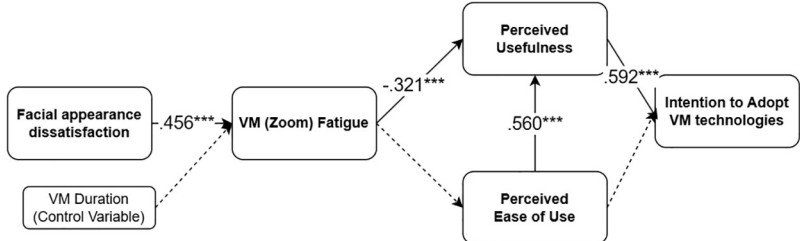

**Fig 8. Model 3 examined for the *low* impression management feature use group.**

effects do not differ significantly depending on the level of impression management feature use. See Figs 7 and 8.

## Discussion

Despite the promise of VMs to make workplaces more efficient and productive, widespread adoption is hindered by the myriad barriers to daily use, stemming in part from psychological and physiological stress of VMs. This study adapts the Technology Acceptance Model (TAM) to examine serial mediating effects of facial dissatisfaction and VM fatigue along with the moderating role of impression management behaviors in VMs to predict behavioral intention to adopt VM technology. Our results demonstrate that as individuals' facial appearance dissatisfaction increases, they experience more VM fatigue, subsequently leading to higher impression management feature use during VMs. VM fatigue induced by facial dissatisfaction, in turn, diminishes perceived usefulness, ultimately affecting intention to adopt VM platforms in workplace meetings. Our findings reveal that VM fatigue is negatively correlated with perceived usefulness, but not with perceived ease of use, contrary to our initial expectations. These findings suggest that the perceived usefulness of VM and affiliated impression management is driven by societal and psychological factors, which ultimately impact decisions regarding adoption. This study utilizes TAM to demonstrate that increased VM fatigue associated with facial dissatisfaction forecasts intention to adopt VM technology through a negative relationship with perceived usefulness. This study did not find a moderating effect of impression management in forecasting the behavioral intention.

The mediating role of VM fatigue in forecasting the association between facial dissatisfaction and impression management feature use in VM supports the reasoning that the motivation for engaging in VM impression management behaviors may relieve the stress associated with concerns regarding self-appearance. This also suggests that negative evaluations of self-appearance can be a substantial cause for VM fatigue. Individuals who suffer from the malady

are more likely to use features that can enhance their appearance to alleviate stress stemming from excessive self-scrutinization and fatigue. Our findings are consistent with previous research regarding facial appearance concerns and VM use [2,3,77]. We replicated and extended the known relationship between facial dissatisfaction and VM fatigue by adding impression management feature use as a final outcome.

## Theoretical implications from the UG perspective: Understanding workers' socio-psychological needs in VM interactions

Our results revealed that facial appearance dissatisfaction is related to impression management feature use through VM fatigue. This aligns with research indicating that individuals who are unhappy with their appearance tend to manipulate their pictures in social networking environments [68]. Our result extends this understanding to real-time communication in the virtual workplace context, providing notable implications from the UG perspective. Namely, individuals use the available components of virtual platforms to fulfill their communication objectives and needs in the workplace and impression management is an important facet to satisfy these gratifications.

People have a desire or need to improve their appearance in the workplace—to engage in impression management—and this can contribute to meeting satisfaction and decreased stress levels in meeting interactions [30]. Additionally, consistent with the UG perspective that relaxation from anxious situations is a key motivator for media technology selection, our study suggests that people with more VM fatigue due to facial appearance dissatisfaction engage in appearance-related impression management behaviors [83,84]. Consequently, our findings imply that current impression management tools (e.g., touch-up, filters, and avatars) afforded by VM technologies may provide uses which gratify workers' needs for self-presentation within workplace communication [81], potentially influencing their professional success [56,125]. Given these theoretical implications derived from the UG perspective, future research on the use of workplace VM technologies should consider workers' psychological and social needs in the context of meeting interactions. For instance, as individual behaviors are often influenced by the identity characteristics of their avatars [126], future research should explore if VM technologies for professional environments that incorporate such self-representational features can reduce fatigue and enhance workers' confidence and efficacy in meeting interactions.

## Theoretical implications: Extending CMC theories, SIP theory and the Hyperpersonal model to virtual meetings

Our finding that facial dissatisfaction is positively related to the use of impression management features through the mediating role of VM fatigue suggests that users may want to overcome the challenges related to impression management in VMs by engaging in selective self-presentation. This aligns with the principles of SIP theory and the hyperpersonal model, which explain how individuals adapt to their digital communication environments, particularly in cue-sparse contexts such as text-based CMC [69,70]. However, the nature of adaptation differs in videoconferencing platforms, which do not exhibit the same level of nonverbal cue scarcity as text-based CMC platforms. Rather than relying on present verbal communication to compensate for the absence of nonverbal cues, as in text-based CMC [69], individuals in videoconferencing meetings must adapt to nonverbal cues which are presented in constrained ways which do not reflect the norms of face-to-face communication [5]. For example, looking at the person to your right in an in-person meeting can act as a nonverbal turn-taking cue. The same physical behavior does not present the same nonverbal information in videoconferencing

meeting environments, where the direction of your head usually does not signal the direction of your attention.

Just as research using SIP theory and the Hyperpersonal model find that people use available verbal cues in text-based communication environments to fill in for the dearth of normative verbal cues [69,70], our findings suggest that people utilize features such as touch-up tools, filters, and avatars to present themselves as they like [5]. VMs across many modality types, from videoconferencing to immersive environments (e.g., VR, MR, or XR), provide more multimodal communication compared to traditional text-based CMC environments (e.g., email, chat) in the workplace, but are still limited in the cues they present compared to face-to-face communication. Consequently, our study highlights the potential applicability of CMC theories, including SIP theory and the Hyperpersonal Model, to workplace VM contexts. This extends the traditional focus of these text-based CMC perspectives to the multimodal nature of VMs, aligning with suggestions from CMC scholars regarding understanding the interplay of multimodal meeting interactions in information processing [71]. For example, future research can investigate how individuals engage in impression management within embodied avatars in immersive environments in ways that yield positive meeting outcomes comparable to face-to-face meeting interactions.

## Theoretical implications: VM fatigue as a barrier to technology adoption via PU

The significant serial mediation model in our study indicates that negative experiences (i.e., fatigue) with VM technology diminish PU and thus can hamper workers' productivity and interfere with effective future use. Drawing on TAM, our study also suggests that along with these negative experiences, mirror anxiety induced by extended self monitoring can hinder adoption and consistent use of the technology. This phenomenon can be seen in workplace contexts where overuse of a technology casts a pall over a platform that would otherwise be beneficial to users. VM fatigue contains multiple dimensions encompassing both emotional and social exhaustion, both of which are caused by various interconnected social and psychological factors—not merely software or hardware [1,24]. Consequently, social and psychological dynamics surrounding VM environments should be taken into account when considering professional use of these platforms. Technologies interact with a variety of social layers within and outside of organizations [127], necessitating the design of VM platforms that effectively address user needs, desires, and behaviors related to VMs. Given that work meetings entail information sharing, collaboration, and impression management [30,58], VM technologies should incorporate important facets of workplace communication and impression management behaviors. In other words, VMs can encompass many parts of workplace communication which can ultimately cause worker dissatisfaction. For example, nonverbal communication is a substantial aspect of impression management and workplace communication [73]. Given that hindrances to nonverbal communication potentially elicit VM fatigue [5], future work should investigate the role of enhancing nonverbal communication to mitigate VM fatigue.

Contrary to what we anticipated, the link between VM fatigue and PEOU was not significant. This is somewhat inconsistent with a previous finding that technostress was negatively linked to PEOU in remote teaching [103]. The lack of an association between VM fatigue and PEOU in the present study may suggest that hindrances to ease of use—such as complex aspects of the interface while signing into meetings—are not meaningful contributors to VM fatigue during the meeting. Furthermore, this may indicate that VM fatigue is more of a psychological rather than a technological issue [128], which could explain the discrepancy

between the previous study and our results. Future research should further investigate whether the psychological facets of technology-mediated communication are significantly related to PEOU.

Additionally, the lack of support for the moderating role of impression management feature use may suggest that facial appearance dissatisfaction and VM fatigue relate to future usage intentions regardless of whether users engage in impression management. However, we should interpret these findings with caution, as it is common for Wald tests to erroneously fail in identifying significant differences (i.e., Type II errors) in small datasets [129]. Furthermore, the serial mediation effect of VM fatigue and PU in the association between facial appearance dissatisfaction and the intention to adopt the VM technology was marginal for the high impression management feature use group, while it was significant for the low impression management feature use group. This implies that the use of impression management features may have the potential to mitigate the negative impact of VM fatigue in future use intentions. Moreover, the appearance-related impression management features prevalent in present VM platforms—such as touch up tools, voice/video filters, and avatars—may not alleviate the negative impact of VM fatigue to a sufficient degree. Future research should identify users' needs and preferences regarding impression management behaviors in workplace meetings and conduct experimental studies with features which provide significant control over impression management to investigate whether such features indeed mitigate the negative effects of VM fatigue, thereby fostering the adoption of VMs.

## Practical implications

Our results demonstrate that individuals experiencing higher levels of VM fatigue are less likely to adopt VM technologies for work meetings. This suggests that organizations and management leaders should implement strategies to reduce fatigue during VMs to facilitate positive meeting interactions, ultimately fostering a greater intention to use VM technologies in the workplace. Additionally, our finding that individuals experiencing VM fatigue perceive VM technologies as less useful highlights a potential impact on productivity. In other words, the usefulness of the technology may be undermined, regardless of impression management behaviors related to physical appearance in VMs. At the same time, our study reveals that individuals experiencing higher levels of VM fatigue are more likely to use impression management features during workplace meeting interactions. This aligns with research indicating that individuals with greater impression management concerns and higher VM fatigue are more motivated to adopt avatar customization options in metaverse workplaces [85]. These findings suggest that organizational cultures should remain open and flexible in allowing workers to utilize VM features such as avatars to address their self-presentational concerns. This suggestion may intuitively align with contemporary office cultures, such as in Silicon Valley and "new economy" companies, which increasingly incorporate playful and informal activities into office spaces [130–132]. While playful decorations and games (e.g., murals, pool/foosball tables) in physical work meeting spaces are not uncommon, they tend to be starkly absent in VM software.

However, striking a balance between professionalism and playfulness is difficult. For instance, avatars on platforms like Zoom tend to be cartoonish rather than formal or realistic, limiting their effectiveness in addressing impression management concerns in professional meeting contexts [133]. This may partially explain why our study was unable to support the expectation that the use of impression management features would moderate the models predicting VM adoption intention. This might imply that impression management features should be designed in VMs with organizational norms in mind to help participants feel

comfortable and supported in achieving positive meeting experiences. Such features could alleviate concerns about facial appearance and foster positive meeting interactions, especially when applied in immersive, embodied environments [e.g., 134]. Future experimental research should explore the causal effect of these features on reducing VM fatigue and their causal influence on technology adoption and positive meeting interactions. This line of inquiry would enable organizations and management leaders to adopt practices that address workers' self-presentational concerns and needs, ultimately enhancing virtual meeting experiences and supporting technology adoption.

Altogether, the work highlights the complexity that comes from the broader use of VM in organizations and society. These platforms deliver a valuable service—synchronous videoconferencing—that respond to and influence broader societal and psychological structures of normative office culture. The lens of impression management helps us understand how this intricate relationship between the technology and these norms engenders a need for regular adjustment and adaptation by both platform providers and users.

## Limitations and future directions

Our study has several limitations that also present opportunities for future research. First, our participants were recruited solely from the U.S., which limits the generalizability of our findings. While our sample included individuals from diverse racial groups, future studies should recruit participants from various cultural contexts to determine whether the identified relationship between facial appearance dissatisfaction, VM fatigue, impression management behaviors, and adoption holds across different cultural settings. Because organizational cultural norms can vary significantly across regions, profession, company size, etc. it is possible that the mechanisms underlying VM fatigue and impression management related to physical appearance manifest differently. For example, expectations regarding facial appearance in the workplace may vary in degree and quality across organizations and cultural contexts. Therefore, future research should explore how workplace appearance-related norms and VM features that enable appearance customization can address self-presentation concerns and support positive meeting interactions across different cultural contexts.

Second, this study employed a cross-sectional survey design, which prevents us from identifying causal relationships among facial appearance dissatisfaction, VM fatigue, impression management behaviors, and intentions to adopt VMs. Future research would benefit from experimental designs to better elucidate the causal relationships between these variables. Experimental studies could provide deeper insights into the mechanisms driving these dynamics and further refine our understanding of the interplay between these factors in the organizational context. Such research would also provide valuable guidance on how VM technologies and practices can be optimized for future virtual interactions, extending beyond workplace settings to include virtual instructional environments.

## Conclusions

Our study provides insights into the intricate psychological and social repercussions entwined with the adoption and use of VM technology within professional settings. Our study highlights the significance of understanding employees' encounters with technology within multifaceted environments where societal and psychological factors converge with both the users and the technology itself. Particularly, our study contributes to understanding the mechanism of VM fatigue, highlighting that VM fatigue is closely related to individual needs and desires for impression management in workplace VMs. We build on CMC theories, particularly the assumptions of the SIP theory and the Hyperpersonal Model, which suggest that people

adaptively harness available cues within communication modalities to facilitate selective self-presentation. Our study demonstrates that users leverage features provided by VM technologies to fulfill their self-presentation needs. By incorporating the TAM and UG perspectives, we highlight that virtual meeting fatigue was related to facial appearance dissatisfaction and reduced perceived usefulness, which impedes technology adoption intent. We suggest that future research should delve deeper into how current VM features facilitate workplace communication in manners that supports the well-being and socio-psychological needs of workers.

## Supporting information

**S1 File. Appendix 1.** Questionnaire Items Used For this Study.
(DOCX)

**S2 File. Data zef adoption.**
(CSV)

## Author Contributions

**Conceptualization:** Chaeyun Lim, Rabindra Ratan.

**Data curation:** Chaeyun Lim.

**Funding acquisition:** Rabindra Ratan, Maxwell Foxman, David Beyea, Alex P. Leith.

**Investigation:** Chaeyun Lim, Rabindra Ratan, Maxwell Foxman, David Beyea.

**Methodology:** Chaeyun Lim.

**Project administration:** Chaeyun Lim, Rabindra Ratan, Maxwell Foxman.

**Supervision:** Rabindra Ratan.

**Validation:** Chaeyun Lim, David Beyea, David Jeong, Alex P. Leith.

**Visualization:** Chaeyun Lim.

**Writing – original draft:** Chaeyun Lim.

**Writing – review & editing:** Chaeyun Lim, Rabindra Ratan, Maxwell Foxman, David Beyea, David Jeong, Alex P. Leith.

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
