## [Decision Letter · Decision Letter 0]

8 Nov 2024

PONE-D-24-42436Examining Attitudes about the Virtual Workplace: Associations between Zoom Fatigue, Impression Management, and Virtual Meeting Adoption IntentPLOS ONE

Dear Dr. Lim,

Thank you for submitting your manuscript to PLOS ONE. After careful consideration, we feel that it has merit but does not fully meet PLOS ONE’s publication criteria as it currently stands. Therefore, we invite you to submit a revised version of the manuscript that addresses the points raised during the review process.

We look forward to receiving your revised manuscript.

Kind regards,

Professor Anis Eliyana,

Academic Editor

PLOS ONE

Journal Requirements:

4. Thank you for stating the following in the Acknowledgments Section of your manuscript: [This material is based upon a study supported by the National Science Foundation under Grant No. 2128803, SES Division.]

Please remove any funding-related text from the manuscript and let us know how you would like to update your Funding Statement. Currently, your Funding Statement reads as follows: [Rabindra Ratan, Maxwell Foxman, David Beyea, Alex P. Leith have received funding from NSF. This material is based upon a study supported by the National Science Foundation under grant FW-HTF-R: Collaborative Research: Virtual Meeting Support for Enhanced Well-Being and Equity for Game Developers NSF Award Nos. 2128746, 2128803, 2128813, and 2128991 from SES Division.

NSF URL: https://www.nsf.gov/]

5. We note that your Data Availability Statement is currently as follows: [All relevant data are within the manuscript and its Supporting Information files.]

**Additional Editor Comments:**

Based on the reviewers' evaluation, your manuscript has the potential to be published in PLOS ONE after **major revisions**. The reviewers emphasized the importance of transparency in research methods, especially in the data collection process and mechanisms to address the issue of common method variance. Research methodology is a crucial aspect that serves as a primary judgment point for articles published in PLOS ONE. In addition, the implications of the research need to be better elaborated to clarify the contribution of this study to the related literature and to organizations. Overall, the manuscript received positive feedback from the reviewers, and I am optimistic that you will be able to respond to each point of feedback in the best possible way.

Reviewers' comments:

Reviewer's Responses to Questions

**Comments to the Author**

1. Is the manuscript technically sound, and do the data support the conclusions?

Reviewer #1: Yes

Reviewer #2: Partly

Reviewer #3: Partly

2. Has the statistical analysis been performed appropriately and rigorously? 

Reviewer #1: Yes

Reviewer #2: Yes

Reviewer #3: Yes

3. Have the authors made all data underlying the findings in their manuscript fully available?

Reviewer #1: Yes

Reviewer #2: Yes

Reviewer #3: Yes

4. Is the manuscript presented in an intelligible fashion and written in standard English?

Reviewer #1: Yes

Reviewer #2: Yes

Reviewer #3: Yes

5. Review Comments to the Author

Reviewer #1: Thank you for submitting your paper to PLOS ONE. This is a well-written paper that covers a highly relevant and timely topic. However, there are several areas where improvement is needed to enhance the clarity, depth, and rigor of your study. Please consider addressing the following points:

Research Problematization and Contributions: The introduction should clearly articulate the research problem and specific contributions your study makes to the literature. Presenting these aspects more clearly will strengthen the framing and relevance of your research.

Theoretical Foundation and Hypotheses Development: The theoretical foundation requires further development by incorporating recent and relevant literature. This will provide a stronger basis for your hypotheses. Suggested literature includes studies on technology adoption, such as Adoption of Telecommuting in the Banking Industry: A Technology Acceptance Model Approach (DOI: https://doi.org/10.28945/5023). Integrating these works will enhance the relevance and currency of your theoretical framework.

Methodology: The methodology section can be strengthened by providing more detailed information regarding your sample population, the sampling technique employed, and sample adequacy. Additionally, including the questionnaire form in the appendix would add transparency to your study design and improve its validity.

Analysis: The analysis section is sound, and no major changes are necessary.

Theoretical Implications: While you have presented the practical implications well, the theoretical implications are underdeveloped. Expanding on how your findings contribute to the theoretical understanding of your topic would provide a more balanced discussion of the study's impact.

Reviewer #2: Congratulations to the authors for reaching this stage. The manuscript includes several revision suggestions that should be considered, as follows:

1. Improving Manuscript Coherence: To enhance coherence, it is necessary to elaborate on previous references and research in the Introduction and Hypothesis Development sections (citation numbers 1-99). In the Discussion section, however, new references are largely introduced (citation numbers 111-117) without prior mention. Therefore, it is suggested that the authors incorporate these new references (citation numbers 111-117) in the Introduction and Hypothesis Development sections.

2. Separate Paragraph for Discussion on SIP Theory and U&G Theory: The Discussion section should include a distinct paragraph analyzing how the findings confirm and extend both SIP (Social Information Processing) theory and U&G (Uses and Gratifications) theory.

3. Adding Implications and Limitations: After the Discussion section, an Implications and Limitations section should be added. The Implications should clarify the theoretical and practical contributions of the findings, while the Limitations should indicate that this study is restricted to the U.S. context. Future research could explore other contexts, perspectives, theoretical bases, and factors not addressed in this study.

4. Enhancing Robustness with Method Variance Testing: To improve robustness, it is recommended to conduct tests addressing common method variance and common method bias issues.

5. Consistency in Findings Presentation: For consistency in the presentation of findings, the final two sentences in the Abstract should align with the key points in the Conclusion section. Additionally, the Conclusion should elaborate on the SIP theory.

These adjustments aim to strengthen the coherence, theoretical contribution, and rigor of the manuscript.

Reviewer #3: I would like to congratulate the authors on the successful completion of their research and the writing of this article. However, there are several issues I would like to address and suggest for the improvement of this article:

1. The authors need to provide a more detailed explanation of the data collection process. Four models were tested with each sample, and there are two points that need clarification:

a) Was the data collection for the four models conducted simultaneously or at different times? If at different times, please explicitly explain the time gap between sample collections for each model.

b) Did the respondents who filled out the questionnaires for models 2, 3, and 4 also fill out the questionnaire for model 1?

2. The authors need to add explanations on the procedural steps taken to minimize common method variance.

3. The authors should include an explanation of the statistical measures used to ensure that common method variance is not an issue in this research analysis. This is important considering that data collection for each model was (allegedly) done cross-sectionally and used the same measurement scale for each variable.

4. The authors briefly mention the theoretical implications in the conclusion section. However, they should create a separate section dedicated to theoretical implications, especially focusing on the contribution of this research to the TAM, SIP, and U&G theories.

5. In addition to the theoretical implications, the authors also need to elaborate on the managerial implications aimed at organizations.

6. The limitations of this research need to be stated explicitly, especially regarding aspects of the research methods. By openly acknowledging the existing limitations, the authors can provide constructive recommendations for future research.

7. I hope the authors can provide more meaningful and fruitful recommendations for future research. This would enhance the value of the article by offering clear directions for subsequent studies in this field.

6. PLOS authors have the option to publish the peer review history of their article (what does this mean?). If published, this will include your full peer review and any attached files.

Reviewer #1: No

Reviewer #2: **Yes: **Alvin Permana Emur

Reviewer #3: No

---

## [Author Response · Author response to Decision Letter 0]

20 Dec 2024

1. I have made the code available on https://github.com/cy-dgl/zef_adoption.git

2. This is the funding information (statement):

This material is based upon a study supported by the National Science Foundation under grant FW-HTF-R: Collaborative Research: Virtual Meeting Support for Enhanced Well-Being and Equity for Game Developers, awarded by the SES Division, NSF Award Numbers 2128746 (Maxwell Foxman), 2128803 (Rabindra Ratan), 2128813 (Alex P. Leith), and 2128991 (David Beyea). We would also like to thank the AT&T endowment to the Department of Media & Information at Michigan State University, which supports Dr. Ratan’s Endowed Chair position.

---

## [Decision Letter · Decision Letter 1]

29 Dec 2024

Examining Attitudes about the Virtual Workplace: Associations between Zoom Fatigue, Impression Management, and Virtual Meeting Adoption Intent

PONE-D-24-42436R1

Dear Dr. Lim,

We’re pleased to inform you that your manuscript has been judged scientifically suitable for publication and will be formally accepted for publication once it meets all outstanding technical requirements.

Kind regards,

Professor Anis Eliyana

Academic Editor

PLOS ONE

Additional Editor Comments (optional):

The reviewers appreciate your diligence in improving the quality of the revised manuscript. Consistent with the reviewers' opinions, I assess that your manuscript demonstrates a robust methodology and meets the ethical standards, which are the primary criteria of PLoS ONE. Furthermore, your research is highly relevant to contemporary issues, providing outstanding theoretical contributions. Additionally, it benefits organizations that extensively utilize information technology in their operations. Therefore, I am pleased to inform you that your manuscript is now suitable for publication in PLoS ONE.

Reviewers' comments:

Reviewer's Responses to Questions

**Comments to the Author**

1. If the authors have adequately addressed your comments raised in a previous round of review and you feel that this manuscript is now acceptable for publication, you may indicate that here to bypass the “Comments to the Author” section, enter your conflict of interest statement in the “Confidential to Editor” section, and submit your "Accept" recommendation.

Reviewer #1: All comments have been addressed

Reviewer #2: All comments have been addressed

Reviewer #3: All comments have been addressed

2. Is the manuscript technically sound, and do the data support the conclusions?

Reviewer #1: Yes

Reviewer #2: Yes

Reviewer #3: Yes

3. Has the statistical analysis been performed appropriately and rigorously? 

Reviewer #1: Yes

Reviewer #2: Yes

Reviewer #3: Yes

4. Have the authors made all data underlying the findings in their manuscript fully available?

Reviewer #1: (No Response)

Reviewer #2: Yes

Reviewer #3: Yes

5. Is the manuscript presented in an intelligible fashion and written in standard English?

Reviewer #1: Yes

Reviewer #2: Yes

Reviewer #3: Yes

6. Review Comments to the Author

Reviewer #1: Dear Authors,

Thank you for resubmitting the revised version of your paper, You have done a great job in addressing the reviewers' comments. The paper is ready for publication.

Regards,

Reviewer #2: (No Response)

Reviewer #3: Dear Authors,

I greatly appreciate your dedication in addressing each of the reviewers' suggestions. With the remarkable improvements made, the manuscript is now suitable for publication.

7. PLOS authors have the option to publish the peer review history of their article (what does this mean?). If published, this will include your full peer review and any attached files.

Reviewer #1: No

Reviewer #2: **Yes: **Alvin Permana Emur

Reviewer #3: **Yes: **Andika Setia Pratama

---

## [Editor Report · Acceptance letter]

2 Jan 2025

PONE-D-24-42436R1 

PLOS ONE

Dear Dr. Lim, 

I'm pleased to inform you that your manuscript has been deemed suitable for publication in PLOS ONE. Congratulations! Your manuscript is now being handed over to our production team.

Kind regards, 

on behalf of

Professor Anis Eliyana 

Academic Editor

PLOS ONE